# Treatments for HBV: A Glimpse into the Future

**DOI:** 10.3390/v13091767

**Published:** 2021-09-04

**Authors:** Alessandra Bartoli, Filippo Gabrielli, Andrea Tassi, Carmela Cursaro, Ambra Pinelli, Pietro Andreone

**Affiliations:** 1Department of Medical and Surgical Sciences, Division of Internal Medicine, Maternal-Infantile and Adult, University of Modena and Reggio Emilia, 41126 Modena, Italy; alessandra.bartoli25@yahoo.com (A.B.); judeslash1@gmail.com (F.G.); andreatassi.at@gmail.com (A.T.); carmela.cursaro@virgilio.it (C.C.); ambra.pinelli@gmail.com (A.P.); 2Postgraduate School of Allergy and Clinical Immunology, University of Modena and Reggio Emilia, 41126 Modena, Italy; 3Postgraduate School of Internal Medicine, University of Modena and Reggio Emilia, 41126 Modena, Italy

**Keywords:** HBV, immunotherapy, direct antiviral agents, gene therapy, therapeutic vaccines

## Abstract

The hepatitis B virus is responsible for most of the chronic liver disease and liver cancer worldwide. As actual therapeutic strategies have had little success in eradicating the virus from hepatocytes, and as lifelong treatment is often required, new drugs targeting the various phases of the hepatitis B virus (HBV) lifecycle are currently under investigation. In this review, we provide an overview of potential future treatments for HBV.

## 1. Introduction

Hepatitis B virus (HBV) is a major cause of chronic liver disease: more than 257 million people worldwide have chronic hepatitis B (CHB), and 600,000 to one million people die annually due to CHB complications [1]. Available treatments suppress viral replication, but are not able to eradicate the virus from hepatocytes, largely due to the persistence of viral covalently closed circular deoxyribonucleic acid (cccDNA) and the incapacity of the immune system to establish an adequate immune response [2]. This results in a necessity to prolong treatments with nucleos(t)ide analogues (NA) for life; however, this does not eliminate the risk of developing liver cancer [3]. More profound knowledge of the HBV lifecycle has favored pharmacological research, and new drugs against specific targets have been developed or are under investigation [4].

We report a non-systematic review, organized using the following electronic sources: PubMed, EMBASE, Google Scholar, Ovid, Scopus, register of clinical trials (ClinicalTrials.gov), the Cochrane controlled trials register, and Web of Science. Research articles and oral communications presented during the European Association for the Study of the Liver (EASL) and American Association for the Study of Liver Diseases (AASLD) congresses during the last 10 years were also considered. We used the following main search terms: “HBV and new therapies”, “HBV replicative cycle”, “HBV and ongoing clinical trials”, “HBV new vaccines”, HBV and immunotherapy”, and “HBV and genic therapy”. We excluded articles with full text not available, not in English language, book chapters, abstracts, case reports, and articles published before 1990. Finally, we considered references cited among the articles analyzed in the first search.

## 2. Hepatitis B Virus Replication Cycle

HBV entry into hepatocytes is mediated by the interaction between hepatitis B surface antigen (HBsAg) and sodium taurocholate cotransporting polypeptide (NTCP), which is specifically expressed on hepatocytes and functions to uptake bile salts into hepatocytes [5,6]. The tyrosine kinase receptor epidermal growth factor receptor (EGFR) triggers the viral particle internalization through its interaction with NTCP [7,8]. Next, the nucleocapsid is released into the cytoplasm and heads toward the nucleus in an active microtubule-dependent manner [9], then enters the nucleus through the nuclear pore complex [10]. In the nucleus, the relaxed circular DNA (rcDNA) released from viral capsids is modified to covalently closed circular DNA (cccDNA) by host polymerases, ligases, and topoisomerase, and persists episomically in the nucleus of infected hepatocytes [11,12,13,14]. cccDNA is relatively stable in quiescent hepatocytes, and the mechanisms that regulate its stability are complex and not fully understood but include immune responses and cytokine stimuli [15]. A recent study has postulated that cccDNA has a turnover rate of several months, instead of decades, as previously supposed [16], and its half-life has been estimated to be about 40 days [17,18]. Part of the incoming viral genome integrates into the host cell genome and can act as a template for HBs synthesis [19].

cccDNA works as a template for the transcription of viral pre-genomic ribonucleic acid (pg-RNA) and messenger RNAs (mRNAs) encoding for viral proteins. They include HB core protein (HBc), HB E antigen (HBe), and viral polymerase, which has DNA elongation activity for both reverse transcription and DNA-dependent DNA polymerization of three types of HB surface antigens and HB X protein (HBx) [19]. HBV polymerase binds to pgRNA and is subsequently encapsidated inside 120 core protein dimers, forming a new viral particle. During this process, the polymerase interacts with the epsilon stem loop of the pgRNA, forming a ribonucleoprotein complex that is included in the capsid [20,21]. This encapsidation is facilitated by host chaperones and mediated by RNA-binding motif protein 24 (RMB24) [19]. Once in the capsid, viral polymerase retro-transcribes pgRNA to rcDNA. At this point, the viral capsid can be covered by an HBsAg-containing envelope and exit the cell as an infecting viral particle, or return to the nucleus and reconstitute the cccDNA reservoir [17]. Figure 1 presents the principal phases of the HBV replicative cycle.

In the following section, we review the principal novel therapies in development for the treatment of HBV infection. Table 1 and Table 2 detail the clinical trials assessing these new drugs.

## 3. Direct Antiviral Agents

This class includes entry receptor blockers, RNA interference blockers, core protein inhibitors, and novel polymerase inhibitors, which are expected to expand the therapeutic options, alongside the implementation of the already existing ones. Table 1 details the principal clinical trials evaluating these drugs.

### 3.1. Entry Inhibitors

Bulevirtide (myrcludex B or hepcludex, BVT) is a myristolated lipopeptide of the pre-S1 domain of the HBV l-surface protein, which can irreversibly bind to NTCP, thus blocking the entry mechanism of the virus [22]. The European Medicines Agency (EMA) approved the use of BVT for the treatment of hepatitis D virus (HDV) infection in 2020 [23]. HDV-infected patients treated with BVT and peg-interferon (PEG-IFN) showed a reduction in both HBV and HDV replication without reaching a functional cure [24]. A phase 3 study with BVT alone or in association with PEG-IFN is ongoing [NCT03852719, NCT03852433].

### 3.2. RNA Interference

Gene silencing is mostly caused by small duplex RNAs such as short interfering RNAs (siRNA) and microRNAs (miRNA) [25]. siRNA and miRNA bind to complementary mRNA, triggering its degradation [26]. Once miRNA molecules reach the cytoplasm, one strand of miRNA—the guide sequence—binds the RNA-induced silencing complex (RISC) [27]. At this moment, there are two possibilities: miRNA can partially bind mRNA or it can perfectly bind mRNA. In the first case, miRNA carries RISC to mRNA, causing deadenylation and the subsequent degradation of mRNA. In the second case, the perfect base pairing between the miRNA and mRNA can activate the complex Argonaute 2-RISC, which can directly cleave the mRNA. The second case is commonly used for therapeutic applications involving the creation of artificial miRNA. Artificial sequences are synthetized to reprogram silencing in different steps of the miRNA biosynthesis pathways [25]. Then, upon an adequate bond, Argonaute 2 (Ago-2), a part of RISC, can directly cleave the mRNA. As described previously, pgRNA and mRNA-encoding HBV proteins are fundamental for HBV replication, and the inhibition of pgRNA generation may result in the reduction in HBV virions recycling into the nucleus, therefore reducing the cccDNA pool [17]. An important step in the development of treatments with RNA interference (RNAi) is determining how to deliver these molecules to the hepatocytes. Two strategies are available: small-sized (diameter < 100 nm) lipid nanoparticles (LNPs), which avoid sequestration during circulation, and cross-capillary fenestrations to reach hepatocytes while N-acetylgalactosamine (GalNac/NAG)-conjugated particles are taken up by asialoglycoprotein receptors (ASGPR) on hepatocytes [28].

ARC-520 siRNA is well-tolerated and has been shown to cause a significant reduction in HBsAg in untreated HBeAg-positive patients, but not in HBeAg-negative ones or those treated with nucleos(t)ide analogue drugs [29]. Successive trials have shown a mild reduction in HBsAg in these patients [29].

ARC-521 is a second-generation siRNA capable of interacting with a wider spectrum of HBV transcripts, which has been demonstrated to reduce HBsAg and HBV DNA serum levels [30]. However, as lethal toxicity of a delivery formulation in non-human primates has been shown, trials with ARC-520 and ARC-521 were terminated.

JNJ-3989 is a siRNA conjugated with Gal/NAc, which inhibits all HBV transcripts. In a phase II study, it has demonstrated good efficacy, with a reduction in HBsAg both in HBeAg-positive and -negative patients, and was well-tolerated. A phase 2 study involving JNJ-3989 in association with nucleos(t)ide analogues is ongoing [NCT03982186, NCT04129554].

ARB-1467 is a LPNs-conjugated siRNA, which showed a reduction in HBsAg in half of the patients. The effect was more evident if it was associated and administrated biweekly. Nevertheless, the study was completed in 2017, and the company states that their next step will be its association with ARB-1467 and tenofovir, followed by the addition of pegylated interferon [31].

VIR-2218, a Gal/NAc-conjugated siRNA, has been investigated in a phase II study, in order to assess its safety, tolerability, and antiviral effect; however, no data are available yet [NCT03672188, NCT02826018].

### 3.3. Core Protein Inhibitors

Hepatitis B core protein (HBc) consists of a single 183–185 amino acid chain with N- and C-terminal domains connected by a hinge region [32]. The N-terminal domain contains the capsid assembly domain, while the C-terminal domain regulates the reverse transcription of HBV DNA. HBc can engage human and viral proteins and create a firm protective capsid. HBc assembles viral pg-RNA and DNA polymerases into nucleocapsids, where reverse transcription takes place. HBc can also interact with nuclear pore proteins, in order to release viral DNA in cell nuclei [33]. Some studies have suggested that HBc plays a role in modulating the expression of viral and host gene, and in regulating cccDNA function. In particular Heat shock protein (HSP) 90 facilitates the formation of viral capsid, HSP40 accelerates capsid degradation, and chaperones can interact with HBc, regulating its stability and favoring capsid assembly [34]. HBc could be a good therapeutic target due to its versatility, and several core inhibitors (also known as capsid assembly modulators or core protein allosteric modulators; CpAMs) have been developed. These drugs have exhibited great inhibition of new rcDNA synthesis and cccDNA establishment [26]. There are two classes of CpAMs: the first one is composed of heteroaryl dihydropyridines, which determine a misassembled HBc, impairing the building kinetics; the second is composed of phenylpropenamides, which increase the kinetics of capsid formation, leading to the generation of empty capsids [35].

NVR 3-778 belongs to the second category, and has been shown to reduce both serum HBV-DNA and HBV-RNA in HBeAg-positive patients at a dose greater than 1200 mg/daily, in a phase 1 study. A great reduction in HBV-DNA and HBV-RNA can be achieved in subjects receiving both PEG-IFN and NVR 3-778. In this group, a mean reduction in HBV DNA of 1.97 log_10_ IU/mL was observed, while the mean reduction in HBV RNA was 2.09 log_10_ copies/mL. NVR 3-778 or PEG-IFN alone can cause a reduction in serum levels of HBV DNA of 1.43 log_10_ IU/mL and 1.06 log_10_ IU/mL, respectively. HBV RNA decreased by 1.42 log_10_ copies/mL and 0.89 log_10_ copies/mL in patients receiving NVR 3-778 and PEG-IFN, respectively. The drug was well-tolerated [36].

JNJ-6379 is a class II CpAm capable of binding the core protein, which interferes with the assembly and encapsulation of pre-genomic RNA. Moreover, it seems to reduce cccDNA formation, interfering with the capsid dismantling and blocking the editing of HBV virions. In a phase 1 study, JNJ-6379 reduced mean serum levels of HBV-DNA and HBV-RNA from the baseline by 2.0–2.5 log_10_ IU/mL in 28 days, and 32% of patients reached undetectable HBV-DNA levels after 28 days. No significant decrease in HBsAg serum level was observed [37]. A phase 2 study to evaluate the efficacy, in terms of HBsAg reduction, with various associations of JNJ-3989 and JNJ-6379 and NA and PEG-IFN-alpha2a, is now being initiated [NCT04667104].

RO7049389, a class I CpAM molecule that causes abnormal HBc aggregates, showed a 28 day median HBV DNA decline of 2.7–3.0 log_10_ IU/mL; however, viral rebound was observed after treatment [38].

ABI-H0731 (Vebicorvir) is a class II CpAm that is capable of interfering with HBV core protein oligomerization, thus promoting the assembly of aberrant capsids. ABI-H0731 blocks the insertion of pgRNA inside the capsid and the entrance of the empty capsid into the nucleus, thus depleting the cccDNA pool in vitro. In a 28 day study, it has been reported to decrease HBV-DNA and HBV-RNA by 4 log_10_ IU/mL [39]. In another study [NCT03576066], ABI-H0731 was compared with or without NA, where 100% of HBeAg-negative patients had HBV-DNA undetectable and pgRNA < 20 IU/mL after one year of a combination protocol ABI-H0731/NA regimen [NCT03576066]. In a randomized, placebo-controlled phase 1 trial [NCT02908191], 60 participants with CHB infections were randomized to receive 100 mg, 200 mg, or 300 mg of ABI-H0731 or placebo for 28 days. Mean maximum HBV DNA declines from baseline were 1.7 log_10_ IU/mL, 2.1 log_10_ IU/mL, and 2.8 log_10_ IU/mL in the 100 mg, 200 mg, and 300 mg cohorts, respectively. The authors stated that, without an active immune-mediated necroinflammatory event, hepatocyte and cccDNA reductions were anticipated to take weeks to months and, thus, antigen reduction was not expected to be observed in these short studies [39].

GLS4 is a class I CpAm that can induce aberrant nucleocapsid. A phase 1b study examined the tolerance, pharmacokinetics, and efficacy at different dosages (120 mg or 240 mg) in combination with ritonavir (RTV) compared with entecavir (ETV) for 28 days. The mean declines in level of HBV-DNA were −1.42, −2.13, and −3.5 log_10_ IU/mL, respectively, in the three groups (120 mg + RTV, 240 mg + RTV, or ETV), with more virological relapse in experimental arms on day 40. Eight subjects experienced ALT flares requiring silibinin and glutathione treatment [40].

ABI-H2158 is a second-generation core inhibitor, which has been assessed in a phase 2 investigation combined with NA. In a phase 1 study, 27 patients were randomized into four groups: one group received placebo and the others received increasing doses of the experimental drug (100, 300, or 500 mg) for 14 days. A mean change from the baseline in HBV DNA and pgRNA was experienced in groups receiving ABI-H2158. HBV DNA declined from the baseline by −2.3, −2.5, and −2.7 log_10_ IU/mL in patients who received ABI-H2158 100 mg, 300 mg, and 500 mg, respectively. pgRNA had a mean change from the baseline of −2.1, −2.3, and −2.1 log_10_ U/mL in patients who received ABI-H2158 100 mg, 300 mg, and 500 mg, respectively. No differences in the levels of HBeAg, HBsAg, or HB core-related antigen (HBcrAg) were observed [41].

JNJ-0440 is a class II CpAm that acts in two ways: first, it interferes with capsid assembly kinetics, preventing the encapsidation of pgRNA and blocking HBV replication; second, it interferes with capsid disassembly, resulting in reduced de-novo formation of cccDNA. It has been tested in 20 non-cirrhotic, NA-naïve patients with CHB in a phase 1b study for 28 days, and a mean change in HBV DNA from the baseline of −3.3 log_10_ IU/mL and −0.2 log_10_ IU/mL in HBV-RNA was determined, while no relevant changes were observed in HBsAg levels [42].

### 3.4. Novel Polymerase Inhibitors

Novel polymerase inhibitors under investigation are Tenofovir exalidex and Besifovir dipivoxil maleate. Tenofovir exalidex shows a specific liver trophism due to its new lipid conjugated formulation, reducing bone- and kidney-related adverse effects. Interim data have demonstrated that a 100 mg dose of Tenofovir exalidex resulted in a mean HBV DNA 3.63 log_10_ IU/mL [43,44]. Besifovir, a guanosine analogue, showed no differences in antiviral effect compared to Tenofovir after 48 weeks of treatment. L-carnitine deficiency was observed in the treatment group [39]. Advanced trials are ongoing, especially in South Korea.

## 4. Immunotherapy

Chronic HBV infection is dominated by viral evasion of innate immunity and exhaustion of specific adaptive T-cells. It is also characterized by impaired cytokine production and altered T-cell expansion [45]. Immunotherapeutic approaches targeting the different alterations tailored by HBV to the host immune system are reviewed in this section. Each approach targets different dysfunctional steps of the immune response; in particular, the goals of the drugs reported below involve the restoration of dysfunctional HBV-specific immune response, innate and adaptive, the induction of novel HBV immune responses, and the induction of an effective T-cell response. Table 2 summarizes the principal clinical trials conducted with these drugs.

### 4.1. Pattern Recognition Receptors (PRR) Agonists

#### 4.1.1. Toll Like Receptors 7–8 Ligands

Toll-like receptors (TLRs) are cellular receptors that sense viral and bacterial pathogen-associated molecular patterns (PAMPs) and represent the first line of defense against external micro-organisms. TLR7 and TLR8 agonists determine IFN production, the induction of IFN-stimulating genes (ISGs), and the activation of other signaling pathways such as JAK/STAT [46,47]. TLR ligands have been tested in vitro and in vivo in animal models, where TLR7 and TLR8 ligands were the most efficacious among the whole category. TLR stimulation activates not only resident liver cells and immune liver cells, but also recruits circulating immune cells, determining an antiviral diffuse state that can result in hepatitis flares and fever as side effects [48].

The TLR-7 ligand GS9620, Vesatolimod, has demonstrated a durable inhibition of HBV replication, and, in some in vitro trials, it also had effects on cccDNA [49]. These results have also been obtained in animal models. Combining Vesatolimod with RO7020531 (another TLR-7 agonist) and RO7049389 (an inhibitor of capsid assembly) determined a marked reduction in HBsAg levels and HBV-DNA in a murine model [50]. In a recent study, after 20 weeks of Vesatolimod administration in CHB patients with HBV replication suppressed by NA therapy, no deflection in HBsAg levels was observed [51]; even in association with Tenofovir, it failed to reduce HBsAg levels [52].

GS-9688 (Selgantolimod) is a TLR-8 agonist, which is orally administrated, well-tolerated, and induced a sustained antiviral response in a woodchuck model of CHB [53,54]. GS-9688 was quite effective in vitro in reducing HBV-DNA, HBV-RNA, and HBsAg, and demonstrated some efficacy in vivo in combination with oral antivirals and 5% of the treated patients achieved a ≥1 log_10_ IU/mL decrease in HBsAg levels or HBeAg loss at 24 weeks [55].

#### 4.1.2. Retinoic Acid Inducible Gene-I Activators

Retinoic Acid Inducible Gene-I (RIG-I) is an intracytoplasmic PAMP receptor activated by the interaction with double-stranded viral RNA. Once activated, its signal is transduced, determining the activation of transcriptional factors such as nuclear factor kB (NFkB) and interferon regulator factor 3 (IRF3), leading to the subsequent production of cytokines with antiviral activity. RIG-I activation also seems to recognize the epsilon-encapsidation signal in HBV pgRNA, with consequent production of type III IFNs. In addition, RIG-I inhibits the interaction of epsilon stem loop of pgRNA with HBV polymerase, thus suppressing HBV replication [56].

SB9200 (Inarigivir) is a RIG-I and nucleotide binding oligomerization domain-containing protein 2 (NOD2) activator, which has shown interesting results in animal models. The administration of this drug followed by Entecavir has shown a much more potent viral inhibition [57]. The ACHIEVE study evaluated this drug in 80 naïve non-cirrhotic patients, who were randomized in a 4:1 ratio to receive daily ascending drug doses (max 200 mg/day) or placebo for 12 weeks, followed by 12 weeks of therapy with Tenofovir Disoproxil Fumarate (TDF). Both HBeAg positive and negative patients reached a reduction in HBV-DNA, in a dose-dependent manner. A quantitative HBsAg reduction greater than 0.5log_10_ was observed in 22% of patients at 12 or 24 weeks [58]. The clinical trials testing this experimental drug have been interrupted as a patient enrolled in the phase II b CATALYST trial developed liver failure and died of multiple organ failure [59].

#### 4.1.3. Stimulator of Interferon Genes Activator

Cyclic guanosine monophosphate (GMP)-adenosine monophosphate (AMP) synthase (cGAS) recognizes HBV DNA and activates its adaptor protein, stimulator of interferon genes (STING), which determines IFN stimulated gene 56 (ISG56) expression, with subsequent production of type I IFN and HBV replication inhibition [60]. An agonist of STING, 5,6-dimethylxanthenone-4-acetic acid (DMXAA), has been experimented in mouse models and in human cultured cells, showing a significant reduction in viral nucleic acids and HBsAg without affecting cccDNA. STING activators have been demonstrated to cause, in a mouse model, body weight loss and colitis as side effects, due to immune system activation through IFN [61].

#### 4.1.4. Apolipoprotein B mRNA Editing Enzyme, Catalytic Polypeptide-like3

Apolipoprotein B mRNA editing enzyme, catalytic polypeptide-like3 (APOBEC3) enzymes act through cytidine deamination and edit HBV DNA strands. Heat shock protein 90 (Hsp90) augments APOBEC-3-mediated HBV deamination activity. Recent studies have argued that APOBEC3 through cytidine deamination could activate type I IFN and lymphotoxin activity, leading to cccDNA degradation [62].

#### 4.1.5. Thymosin α1

Zatadin is a 28 amino acid peptide isolated from thymus, which seems to activate both innate and adaptive immunity. This peptide is thought to activate and lead to the differentiation of dendritic cells (DCs) and T-cells by TLR-2 and TLR-9, and promotes the expression of IFN-γ and interleukin-2 (IL-2) [63]. In a clinical trial in CHB patients, thymosin α1 (Tα1) showed fewer side effects compared to IFN-α. In combination with Entecavir (ETV), it led to similar serological, biochemical, and virological effects, even if the combination therapy demonstrated more efficacy in inhibiting hepatocellular carcinoma (HCC) development in CHB patients than ETV alone [64].

### 4.2. Checkpoint Inhibitors

#### 4.2.1. Programmed Death-1 (PD-1)/Programmed Death-Ligand 1 (PDL-1) Inhibitors

These prevent the interaction of inhibitory receptors (PD-1) mainly expressed on the surface of T- and B-cells with their ligands (PD-L1 and PD-L2), which can be found on a large variety of cells including hepatocytes. The bond between PD-1 or PDL-1 and the inhibitors blocks CD8+ T-cells inhibition, determining an improvement in cell proliferation, cytokine secretion, and cytotoxic capability [65]. Among the inhibitory receptors, PD-1 has been demonstrated, from in vitro studies, to determine major inhibitory effects. Most responding cells are effector memory HBV-specific CD8+ T-cells from peripheral blood, while intrahepatic HBV-specific T-cells are usually more exhausted and, for their restoration, an association with the blockade of other inhibitory receptors could be useful [66].

An ex vivo study has shown that the combination of PD-1 agonist and OX-40 (also named cluster of differentiation 134, CD 134) synergically acted and resulted in the activation of HBV-specific CD4+ T-cells, with the consequent production of IFN-γ and IL-21 in HBeAg-negative HBV-infected blood cells [67].

The combination of anti-PD-L1 monoclonal antibodies, a nucleoside analogue, and a therapeutic DNA vaccine led to viral replication suppression and seroconversion in a woodchuck model, obtaining a sustained immunological control of viral infection due to potently enhanced HBV-specific T-cell function, but not with anti-PD-L1 monoclonal antibodies alone [68].

In HBeAg-positive patients, treatment with anti-PD-L1 monoclonal antibodies (MEDI2790) augmented HBV-specific T-cell responses while in a phase Ib study, the PD-1 agonist Nivolumab was found to be safe and well-tolerated with or without association to therapeutic vaccination (GS-4774), and achieved a drop in HBsAg by 0.5 log_10_ at 24 weeks, with sustained HBsAg loss and seroconversion in one patient out of 14 [69,70].

Given the wide variability of in vitro and in vivo test results, it seems reasonable to believe that only some patients could benefit from this treatment option, depending on their T-cell exhaustion degree and on the characterization of other target receptors such as other inhibitory receptors, expressed on cell membranes. The resistance to checkpoint inhibitors could also be due to genetic and epigenetic modifications, which can resist PD-1 blockade. Even if checkpoint inhibitors are relatively safe at low dose, the association of two molecules belonging to the same class or the association of a checkpoint blocker and other drugs from other categories could theoretically enhance T-cell recovery; however, on the other hand, safety issues could be raised, especially in the field of generalized immune activation [71].

#### 4.2.2. Cytotoxic T Lymphocytes-Associated Antigen-4 (CTLA-4) Blockade

CTLA4 is another class of inhibitory receptors. In in vitro studies, the blockade of CTLA-4 receptors determined a restoration of HBV-specific CD8+ T-cells with augmented function and proliferative capacity [72] and inhibited T-reg activity, thus enhancing CD4+ Th in a murine model [73]. A CTLA-4 blocker, Ipilimumab, has been employed in a group of patients with melanoma, among whom some were affected by HCV or HBV chronic infections. In this small sub-group of nine subjects, a reduction in viral load was seen in two [74].

## 5. Therapeutic Vaccination

Therapeutic vaccines differ from their preventive counterpart, in terms of their mechanism of action and the time of administration, when the infection has already become chronic. Therapeutic vaccines aim to activate not only humoral immunity but also effective CD4+ and CD8+ T-cell responses [66]. The most effective therapeutic vaccines, in general, seem to be those based on viral vectors, which can majorly activate specific T-cell responses.

DNA vaccines have been shown to be very safe and, therefore, could be used repeatedly as well as in combination with viral vectors [47,53]. Published results of trials regarding all HBV therapeutic vaccine categories as well as ongoing ones are detailed in Table 3.

### 5.1. HBsAg-preS Vaccines

Prophylactic HBsAg vaccines have not achieved a therapeutic result in chronic HBV patients due to the high circulating levels of HBsAg. Zeng et al. added IL-12 as an adjuvant in an HBsAg vaccine in a mouse model, demonstrating a breaking of tolerance with enhanced HBV-specific CD8+ and CD4+ responses and reduced T-reg cells. Many animals lost HBsAg, and HBcAg was undetectable in mouse hepatocytes [75]. Sequential administration of pre-S1 polypeptide to HBsAg vaccines in HBV-infected mice led to an important immune response with the development of anti-pre-S1 antibodies, which determined HBV virion clearance and HBsAb seroconversion [76].

### 5.2. HBcAg Vaccines

Based on the discovery that, in patients capable of controlling HBV infection, a large number of HBcAg-specific CD8+ T-cells was found, HBcAg vaccines are now considered a promising option for CHB treatment [77].

### 5.3. HBsAg-HBcAg Compound Vaccines

Therapeutic vaccines combining HBsAg and HBcAg have shown strong HBsAg/HBcAg immune responses in animal models, leading to HBsAg serum reduction without known liver side effects in transgenic mice [47]. A candidate vaccine composed of HBsAg, HBcAg, and a saponin-based adjuvant induced a potent humoral and CD8+ response: anti HBsAg titers were >10,000 IU/L in seven of eight animals after a cycle of four vaccinations. HBcAg-positive hepatocytes were also cleared without detectable liver injury [78].

### 5.4. Anti-HBsAg Antibodies

Antibody-mediated immunotherapy has been tested in many clinical and pre-clinical studies without obtaining remarkable results. Zhang et al. developed a monoclonal antibody (mAb E6F6) directed against HBsAg in a CHB mouse model, with notable results consisting of HBsAg and HBV DNA suppression for several weeks. An interesting finding consisted of the observation that HBV-specific T-cells were restored after E6F6 immunotherapy in hydrodynamic injection-based HBV carrier mice [79,80].

### 5.5. DNA Vaccines

DNA vaccines encoding for HBcAg and HBsAg are administrated by intramuscular or intradermal injection, and determine HBsAg and HBcAg expression by local antigen-presenting cells as well as the consequent activation of specific T- and B-cells. In pre-clinical animal studies, better results have been obtained by boosting the immune response with the co-administration of immunostimulant cytokines, NA combination therapy, prime boost immunization regimens, electroporation delivery of injected substances, or adding checkpoint inhibitor therapy [47]. Chuai et al. experimented with a complex vaccination method: they vaccinated four rhesus macaques with three HBV-DNA vaccine doses coding for HBsAg, pre-S1, and HBcAg, followed by two boosts with recombinant viral vector vaccines encoding HBsAg, pre-S1, and HBcAg for priming, and a final boost constituted by a fusion protein composed of HBsAg and pre-S1. The first antibodies to appear were those against pre-S1, followed by anti-HBsAg and -HBcAg antibodies. Humoral and cellular responses were detected for all three components, where the anti-HBcAg T-cell specific response was the strongest and most durable. Further boosting with the fusion protein warranted the immune response against all three proteins after 98 weeks from the first vaccination. This showed that the addition of pre-S1 and HBcAg in therapeutic vaccines can improve the immune response [81].

### 5.6. Vaccines Based on Viral Vectors

Therapeutic vaccines based on viral adeno-vectors obtained good results in mouse and woodchuck models, as adenoviruses have been demonstrated to provide a strong stimulation and activation of T-cell compartments [66]. An experimental vector-based vaccine, TG1050, consisting of a non-replicative adenovirus 5 vector encoding for a fusion protein, comprising a modified core, polymerase, and selected domains of HBV envelope proteins, seemed to be effective in driving a strong T-cell response, with an anti-viral effect in HBV chronically infected mice. This molecule is now under evaluation in a phase Ib study in chronic HBV patients already in treatment with NAs [82]. Another vaccine, based on an inactivated parapoxvirus (iPPVO), AIC649, underwent a phase I trial, showing an increase in IL-1β, IL-6, IL-8, and INF-γ levels. The results indicated that it was well-tolerated [83].

## 6. HBV-Specific T-Cell Therapy

### T-Cell Engineering

This technology has been extended to the HBV field in order to overcome specific CD8+ T-cell exhaustion in CHB patients. Current T-cell replacement therapies use different types of engineered T-cells, targeting specific immunodominant viral epitopes with the purpose of replacing non-functioning endogenous CD8+ cells or to enhance their reduced functions. This strategy may lead to the complete and continuous control of HBV infection by the immune system and its cure in the long-term [84]. The major part of genetically engineered T-cell transfer therapies have been applied in the oncological field, but they have found applications in the fields of autoimmune diseases and, lately, infectious diseases. In the specific HBV case, the discovery of this possible therapeutic approach derived from the observation that bone marrow transplantation, from previously infected HBV donors who had cleared the virus, to CHB patients led to viral clearance [85]. In the same way, liver transplantation of an HBsAg-positive graft in patients with resolved HBV infection resulted in HBV clearance [86].

Two methodologies have been developed consisting of infusing a patient with their autologous T-cells, engineered to express a canonical human leukocyte antigen (HLA) class I restricted T-cell receptor (TCR) or a chimeric antigen receptor (CAR) targeting a specific HBV epitope. In the first case, the patient’s circulating T-lymphocytes are isolated and then expanded and activated in vitro; this process is followed by the redirection of CD8+ T-cell specificity toward HBV epitopes using viral vectors encoding a specific TCR [71,87]. In a patient with CHB and HCC, this methodology has led to a drop in HBsAg production by HCC cells [87]. An important advantage of this method is its specificity: in another study, short HBV-DNA integrated into an HCC cell genome was used as the target of redirected T-cell-specific TCR and, in this way, pulmonary metastases were reduced as they became the specific target of activated CD8+ T-cells [88]. A further evolution, recently experimented in mice, consists of the transient expression of HBV specific T-cell receptors by electroporation of specific mRNAs. The generation of HBV-specific CD8+ T-cells with reduced lytic capacity (reduction in granzyme B and perforin production), but which are capable of activating the cytidine deaminases APOBEC3 in HBV-infected cells, limited viral infection, and also prevented an excessive immune response, protecting against possible liver damage and failure [89].

The CAR approach, instead, includes an HBV antibody-specific fragment with CD28 (a co-stimulatory molecule) and the CD3 zeta intracellular domain. These engineered T-cells can recognize HBV-infected cells independent of the patient’s HLA haplotype. From the experimental studies, the delivery of these chimeric receptors by retroviral vectors to HBV infected cells expressing HBsAg determined cell death and cccDNA elimination when the chimeric antigen receptors used were those against HBsAg [90].

The implementation of these technologies seems difficult due to the large amounts of engineered T-cells and the technical skills needed. Moreover, even if the process seems focused and precise, too strong a T-cell-mediated immune response could determine acute liver damage and failure; on the other hand, operating in an inhibitory environment, the new T-lymphocytes could suffer the same fate as the endogenous ones and run into exhaustion [91].

## 7. Gene Therapy

### 7.1. Designer Nucleases

Nucleases act by cutting HBV double-stranded DNA at a pre-defined site. Once the double-stranded break is created, the DNA repair systems of host cells, as non-homologous end joining (NHEJ) or homology-directed repair machinery, attempt to repair the damage. As NHEJ is prone to error without a donor template, it may relegate cleaved ends and randomly insert or delay nucleotides, resulting in mutations. HBV cccDNA is the perfect candidate for nuclease gene editing due to its episomal minichromosome configuration and narrow sequence plasticity [92].

### 7.2. Zinc Fingers (ZFs)

ZFs are mammalian proteins with transcription factor functions. They are composed of three to six DNA-binding cysteine2 (Cys2)–histidine2 (His2) zinc fingers coupled with a restriction single-strand endonuclease derived from *Flavobacterium okeanokoites* (FokI). As they are monomers cutting single-strand DNA, in order to determine a double-stranded break, ZFs require monomer dimerization and juxtaposition on the target DNA sequence [92]. Due to issues relating to cytotoxicity and difficult construction methods, the advancement of therapeutic application of ZFs has suffered from delays [92].

### 7.3. Transcriptional Activator-like Effector Nucleases (TALENs)

TALENs are dimeric engineered nucleases constituted of a DNA binding protein derived from Xanthomonas bacteria and a FokI endonuclease domain. The TALE protein is composed of 12/20 tandem repeats of 33–35 amino acids. The repeat sequencies are almost identical, except for position 12 and 13 amino acids, called repeat variable residues (RVDs), which are responsible for exact nucleotide recognition and site-specific DNA binding [93]. Chen et al. designed TALENs targeting the HBV DNA conserved regions comprised between polymerase (RNAse H sequence) and C ORFs in liver-derived Huh7 cells. As a result, reductions in viral production of HBeAg, HBsAg, HBcAg, and pgRNA as well as in the cellular cccDNA concentration, was achieved. Combining IFN-α with TALENs, a further reduction in cccDNA concentration has been observed [94]. This approach revealed encouraging results, both in vitro and in animal models, determining a reduction in cccDNA [93,94,95].

### 7.4. Clustered Regularly Interspaced Short Palindromic Repeats (CRISPR)—CRISPR Associated Protein 9 (CAS9)

CRISPR/CAS9 represents the most famous RNA-guided nuclease (RGN), which is now thought to be the most diffuse methodology to inactivate HBV gene expression. These RGNs are normally constituted of a CRISPR RNA (crRNA), which comprehends complementary sequences to a pre-defined targeted DNA sequence and a transactivating crRNA (tracrRNA). The bond between the crRNA and its complementary target DNA determines the activation of Cas9 nuclease and DNA double-stranded cleavage. The cleavage of cccDNA, with the formation of a linear sequence, destabilizes the viral target [93]. Lin et al. were the first to demonstrate the efficacy of CRISPR/Cas9 in splitting intrahepatic HBV DNA and clearing viral replication in vivo [96]. From a next-generation sequencing map, the majority of cccDNA indels induced by CRISPR/Cas9 activation manifested as small deletions [97]. Many other experiments followed, demonstrating the efficacy of CRIPR/Cas9 in inhibiting HBV replication and reducing cccDNA nuclear amounts [98]. A side effect of this technology is high rates of possible non-target cleavage; however, scarce data are available. To obtain better results in viral clearance, or to excise integrated viral DNA, the use of multiple single guide RNAs (sgRNAs) is required, but could also worsen off-target cleavage. These events can be detected using deep sequencing when multiple sgRNAs are exploited. In order to tackle this issue, Cas9 endonucleases can be replaced with engineered nickases, which are more specific as, in order to obtain a double-strand break of targeted DNA, two Cas9 nickases need to match. Another central issue to be improved consists of CRISPR/Cas9 delivery to the hepatocytes [93]. The only data available at present regard animal models and human cell cultures. In order to develop technology suitable for human clinical trials, further advances are needed [98].

### 7.5. Synthetic mRNAs as a Therapy

This field is in continuous development and the objectives that need to be reached are multiple: large-scale production of therapeutics, accurate dose regulation, safety, prevention of side effects derived from off-target cleavage, and efficient and precise cell delivery. Using mRNA instead of DNA to express designer nucleases could, therefore, represent a smart trick. mRNAs are rapidly reproducible, have a short half-life, they cannot recombine with the host genome, and they are correctly delivered to the cytoplasm and not the nucleus of cells [98]. Some studies have evaluated lipid nanoparticles as synthetic mRNA transporters to liver cells [99]. In vitro transcribed mRNA has been utilized to express TALENs, and naturally synthetic mRNAs have also been applied in a CRISP/Cas9 system with encouraging results [98]. Little work has been carried out, so far, in terms of disrupting HBV-DNA through the use of gene editors encoded by synthetic mRNAs; the few studies performed so far have obtained modest results [100].

### 7.6. Epigenetic Gene Silencing

The epigenetic modification of DNA is a natural regulatory mechanism, which represents a host defense against the expression of viral genes. It is determined by chemical modifications of DNA or associated proteins without changing genetic information [93]. Growing evidence has highlighted the importance of epigenetic modifications in HBV cccDNA transcription control and the fact that it could represent a strategic target in CHB treatment [101]. Epigenetic modifications include histone methylation and demethylation, histone acetylation and deacetylation, cccDNA methylation, and cccDNA minichromosome acetylation. The major HBV DNA epigenetic modifiers are histone deacetylases, histone acetyltransferases, lysine methyltransferases, DNA methyltransferases, and protein arginine methyltransferases, without forgetting viral factors such as HBX and core antigen [93]. The hypoacetylation of cccDNA-bound histone3–4 determined low HBV viremia in hepatitis B chronic infected patients, while its hyper-acetylation enhanced viral replication [102]. The methylation of arginine-3 on cccDNA-bound histone4 prevented its bonding with the RNA polymerase and transcription in an arginine methyltransferase5- and HBc-dependent way [101]. The application of this technology could help in reducing the HBV reservoir by stimulating hepatocyte turn-over and preventing cccDNA pool renewal as it accelerates cccDNA deterioration [103]. Despite the modern technologies and the importance of epigenetic modifications for viral gene expression, clues to support the potential of epigenetic modifiers in HBV chronic infection are limited. In fact, few studies have demonstrated the advantages of using an interesting technique: exchanging the nuclease domain of a designer nuclease with an epigenetic modulator. The previously cited ZFs, TALEs, and CRISPR/Cas9 have been modified, in order to enable the epigenetic editing of endogenous genes [104]. Another study targeting HBx promoter reported a reduction in viral mRNA, proteins and viral replication in a cell culture as well as in HBV transgenic mice [105]. Another therapeutic approach involved a transcription repressor, the Krüppel associated box; this was bound with HBV targeting TALE or a dead Cas9, used together with gene-targeting guides, with an observed reduction in serum concentrations of HBsAg and surface and core mRNAs in mouse cell cytoplasm. Animals tolerated this new methodology well and did not present alanine aminotransferase (ALT) flares [106].

## 8. Targeting HBV Core or X Protein to Inactivate cccDNA Function

cccDNA is an episomal minichromosome, whose transcription is regulated by epigenetic factors. Many studies have suggested that HBV core and HBx proteins regulate cccDNA transcription. Another option to disrupt or inactivate cccDNA consists of the interference between these proteins and their binding with it, or the selective inhibition of interactions between viral proteins and host factors necessary to regulate cccDNA transcription [107]. HBV core protein is a fundamental protein in cccDNA homeostasis, as it regulates its transcription. The protein binds CpG islands of cccDNA, preventing methylation and silencing [108]. The cytokine-activated DNA cytosine deaminase APOBEC3A can be recruited to the cccDNA minichromosome through interaction with core protein; this provides evidence of the possibility that core protein can be exploited by the immune system to its advantage—that is, to clear cccDNA [63].

Pharmacological disturbance of core–host and core–core protein interactions could be a key in HBV treatment and, thus, should be carefully characterized. In this respect, there exist many new and experimental drugs to inhibit and disrupt capsid assembly, which could be useful in disturbing core–cccDNA interactions.

HBx is, among all its other functions, a trans-activating protein capable of inducing the transcription of the cccDNA minichromosome. It is responsible for epigenetic modifications on cccDNA, by histone acetylation, demethylation, and inhibition of cccDNA transcriptional repressors [107]. HBx protein also indirectly degrades the structural maintenance of chromosomes 5/6 (Smc5/6) complex, which act as cccDNA transcriptional suppressors [109].

## 9. Immune Modulation for cccDNA Attenuation

The only licensed immunomodulatory drugs in chronic HBV infection are IFN-α and its pegylated form, which have only shown partial efficacy, accompanied by high amounts of adverse effects (mostly related to the systemic bioavailability) and scarce tolerability. IFN-α seems to act on both the innate and adaptive immune response, showing some degrees of efficacy in indirectly disrupting cccDNA by reducing its histone 3 lysin 9 (H3K9) and histone 3 lysine 27 (H3K27) acetylation. Efforts have been made to develop a gene therapy capable of inducing a durable activation of immune modulators, in order to permanently attenuate cccDNA expression. The liver-localized expression of IFN-α-specific sequences could improve the drug efficacy, while reducing side effects. Using different viral vectors, diverse research groups have demonstrated the possibility of transporting IFN-α codifying sequences into the livers of hepatitis B chronically infected mice and woodchucks, then to locally express them [93]. In a woodchuck hepatitis virus (WHV) model, Fiedler et al. demonstrated that the liver localized expression of IFN-α was associated with a reduction in intrahepatic WHV replication and a decrease in WHV DNA in serum, accompanied by an acceptable tolerability profile [110]. Other researchers have experimented with IL-12 and IL-15 plus IFN-α combinations, obtaining excellent results. After these optimistic findings in mice, the authors decided to test their combination strategy on samples from CHB patients, proving a restoration of HBV-specific CD8+ specific cells. Human clinical trials have not been performed yet [93].

## 10. Farnesoid-X Receptor Agonists

The bile acid-regulating farnesoid-X receptor (FXR) is part of the more complex NTCP receptor, which mediates the entry of HBV into hepatocytes. FXR activation inhibits HBV replication [26]. At present, there are five active studies, of which one is in phase II. This study has suggested, on the basis of pre-clinical data, the potential inhibition of HBsAg and HBeAg production [NCT04365933].

## 11. Cellular Inhibitors of Apoptosis Proteins (cIAP) Inhibitors

cIAPs counteract the pro-apoptotic effects of TNF-α. Recent studies have proven that Birinapant, a cIAP inhibitor, leads to HBV clearance in hydrodynamically infected mice, enhancing apoptosis through TNF-α and activating a CD4+-specific T-cell response. Birinapant has also recently been studied in association with a multi-drug resistance protein-1 efflux pump inhibitor, Zosuquidar, in order to augment its cell concentration. Compared to Birinapant alone, the combination was more efficient in controlling HBV-DNA and HBsAg kinetics as well as in terms of targeting HBV-infected cells, in a mouse model of HBV infection [111]. Another cIAP inhibitor (APC-1387) is currently under investigation in clinical trials (see Table 2; NCT04568265).

## 12. Nucleic Acid Polymers

Nucleic acid polymers (NAP) are phosphorothioate oligonucleotides that perform their action in a sequence length- and phosphorothioate-dependent manner. A phosphorothioate is a compound containing a sulfur in place of one of the non-bridging oxygen atoms; this creates a negatively charged molecule. The presence of sulfur atoms, instead of oxygen, causes weaker electronegativity and, so, these compounds have a major hydrophobic nature. Thus, when it is exposed to a local hydrophobic environment, the free electron in the non-bridging sulfur can transit from a charged to uncharged state. This can be exploited by phosphorothioate oligonucleotides, in order to localize the hydrophobic property in a part of this molecule, thus allowing it to interact with hydrophobic proteins [112].

We can resume this process by saying that NAP activity is led by the interaction of large amphipathic protein domains, which are important for viral replication.

NAPs are antiviral agents used for different infections such as HIV as well as for HBV and HDV. For HBV, NAPs are able to block the replenishment of HBsAg in circulation and block the release of HBsAg from infected hepatocytes, thus helping in host-mediated clearance [113]. However, the exact mechanism remains unclear. REP 2139 is one of the first NAPs tested for both HBV and HDV. CHB-infected patients with HBeAg negativity, anti-hepatitis D antigen (HDAg) positivity, and HDV RNA positivity with HBsAg concentrations of more than 1000 IU/mL were enrolled in an REP 301 trial. Patients received intravenous REP 2139 once weekly for 15 weeks, then a combination of REP 2139 and PEG INF alfa 2a for another 15 weeks, followed by PEG INF alfa 2a for the last 33 weeks [NCT02233075]. An additional follow-up (at one year) is still ongoing [NCT02876419]. A total of 12 patients were enrolled. HBsAg loss, high anti-HBs titers, and suppressed HBV DNA was reached in five out of 12 patients, with normalization of aminotransferase in nine of 12 patients. Transient elevations in ALT and AST (10 × upper limit of normality, UPN) were observed, especially with the introduction of PEG INF alfa 2 a, as a consequence of the antiviral response [114].

Two trials have evaluated the safety, tolerance, and efficacy of REP 2055 (REP 101; NCT02646163) and a modified version of REP 2055 with a novel calcium chelate complex formulation, REP 2139-Ca (REP 102; NCT02646189) in non-cirrhotic CHB naïve to treatment patients. In REP 101, seven of eight patients experienced a reduction in serum of HBsAg, varying from 2.03–7.2 log_10_ IU/mL as well as the appearance of anti-HBs > 10 mU/mL in all patients. All seven responding patients experimented a reduction in HBeAg or HBeAg seroconversion. Gum bleeding, fevers, and headaches were observed with a transient elevation of ALT and AST. In REP 102, nine of 12 participants showed reductions in serum HBsAg, varying by 2.79–7.1 log_10_ from baseline values, and a decline in serum HBV DNA ranging from 3–12 log_10_ was observed. Three participants in this experiment showed HBsAg loss during monotherapy. Five patients presented anti-HBs > 10 mIU/mL during REP 2139-Ca monotherapy [115].

## 13. Conclusions

The possible strategies to eradicate HBV infection are various. While the research regarding many classes of DAA consists of advanced clinical trials, immunotherapies, and therapeutic vaccines are still in previous development phases, although with promising results. Furthermore, new strategies—including epigenetic gene silencing, T-cell engineering, HBx targeting, and others—are in the initial phases of investigation.

The rising consensus in the field is that combination therapy may be the key to achieving future therapeutic goals. Considering the different steps in viral replication and the multiple alterations in the host immune system due to HBV infection, the options available for combination therapy are considerable.

NAs could be associated with different DAAs such as entry inhibitors, for instance, Myrcludex B. In this case, even if the combined effects may be stronger than that of the single molecule, the pool of already infected cells would not be affected and, so, a functional cure would be hard to reach [116]. Another association option could be constituted by NAs and cccDNA inhibitors. Their combination could strongly determine a drop in cccDNA and in HBV DNA, leading to a possible complete cure. cccDNA inhibitors are a heterogeneous class of different molecules. CRISPR/Cas9 presents different obstacles including delivery issues and risk of off-target effects on the host genome [98]. Epigenetic modifiers, on the other hand, also present specificity issues as they may alter chromatin regulation [93].

A third option could be represented by the association of NAs with CpAMs; early study data have shown the high efficacy and synergistic effects of CpAM when combined with NAs or IFN in in vitro models; however, considering the viral cycle steps on which this molecule association acts, a long therapy should be performed, in order to obtain cccDNA degradation effects [26], [NCT04667104]. NAs could also be associated with siRNA, which have shown good results in clinical trials thus far (see Table 1) [29]. There are, however, safety and delivery issues and risk of pharmacological resistances [117].

The associations of different immunotherapies have also been postulated. In order to achieve a specific immune control through exhausted T-cell restoration, checkpoint inhibitors, engineered T-cells, and therapeutic vaccinations have been developed. Each approach shows important difficulties, for example, autoimmune phenomena as side effects for the first, technical issues for the second, and disappointing results in human studies for the third [116]. The association between checkpoint inhibitors and therapeutic vaccines is, among the others, very interesting: on one hand, there is a strong restoration of exhausted T-cells; on the other hand, the now effective adaptive immunity is activated against a specific HBV epitope [63]. Despite the potential synergistic effect, issues have been raised in the field of excessive immune response, with possible immune-mediated hepatitis as a side effect [118].

Combining drugs targeting the virus itself (DAAs) with immunotherapy may represent the most rational approach to reach a functional cure, leading to undetectable HBsAg levels and suppressed HBV DNA [24]. Different attempts associating NAs and therapeutic vaccinations, TLR-7/8, anti-PD-1 monoclonal antibodies, FxR agonists, and cIAP inhibitors are now ongoing (see Table 2). Advanced technologies such as TALENs, CRISPR/Cas9, and synthetic RNAs are still in a developmental phase, but represent interesting therapeutic options. In conclusion, the deeper knowledge and the more recent and developed technologies have contributed, in a couple of decades, toward many new possible pharmacological approaches for the treatment of one of the historically most difficult infections to treat. Many trials with different agents are now proceeding, which will hopefully lead to encouraging results. Hopefully, in the next ten years, we will experience a complete revolution in the treatment paradigm of HBV infection.

## Figures and Tables

**Figure 1 viruses-13-01767-f001:**
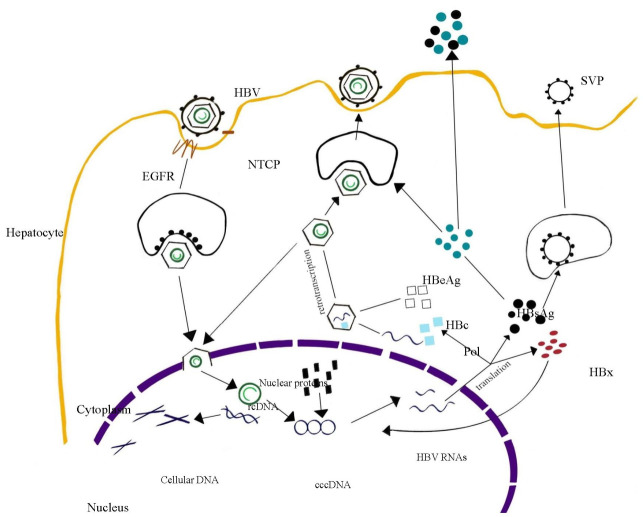
The replicative cycle of hepatitis B virus (HBV). Details about the various phases of this process are described in the text. HBV: hepatitis B virus; EGFR: epithelial growth factor receptor; NTCP: sodium/taurocholate cotransporting polypeptide; rcDNA: relaxed circular deoxyribonucleic acid; cccDNA: covalently closed circular DNA; HBV RNAs: HBV ribonucleic acids; Pol: viral polymerase; HBx: hepatitis B protein x; HBc: hepatitis B core protein; HBeAg: hepatitis B E antigen; HBsAg: hepatitis B surface antigen; SVP: small vesicle particle.

**Table 1 viruses-13-01767-t001:** Principal clinical trials evaluating new direct antiviral agents. PEG-INF: pegylated interferon alpha; Myr: Myrcludex; ETV: entecavir; CHB: chronic hepatitis B; CHD: chronic hepatitis D; TDF: tenofovir disoproxil fumarate; TAF: tenofovir alafenamide; RNA: ribonucleic acid; DNA: deoxyribonucleic acid; BSV: besifovir; HBsAg: hepatitis B surface antigen; HBeAg: hepatitis B e antigen; HBV: hepatitis B virus; HDV: hepatitis D virus; ALT: alanine aminotransferase; AST: aspartate aminotransferase; pgRNA: pre-genomic ribonucleic acid.

Entry Inhibitors	Reference	Study Type	Results	Adverse Events, Limitations/Comment
Bulevirtide (Myrcludex B, Myr)	ClinicalTrials.gov Identifier: NCT02888106	Phase 2 randomized, comparative, parallel-arm study in combination with peginterferon alfa-2a versus peginterferon alfa-2a alone in patients with CHB with delta-agent.	HBsAg response defined as ≥1 log_10_ decline or negativity at 24 weeks after the end of treatment was noted in 46.7% of the patients treated with bulevirtide 2 mg plus PEG-IFN-α and in 20% receiving bulevirtide 5 mg plus PEG-IFN-α. RNA HDV negativization was reached majorly in groups treated with Myrcludex.	Total bile acids increased, fever, flu like symptoms, and neutropenia. Phase 2 study with a small study population
	ClinicalTrials.gov Identifier: NCT03546621	Phase 2 multi-center, open-label, randomized clinical trial on Myr in combination with tenofovir, compared to tenofovir alone for the suppression of HBV replication in patients with CHD.	HDV RNA negativization or decrease were statistically superior in groups receiving Myr + tenofovir	Blood and lymphatic system disorders, anemia, Total bile acids increase, ALT and AST increase, and nausea. Small study population
	ClinicalTrials.gov Identifier: NCT02881008	A randomized, open-label multi-center clinical trial considering Myr at four different doses vs. entecavir.	No differences in response between Myr- or ETV-treated groups	Eosinophilia, leukopenia, abdominal pain (upper), asthenia. Small study population
**RNA Interference**	**Reference**	**Study Type**	**Results**	**Adverse Events, Limitations/Comment**
ARC-520	ClinicalTrials.gov Identifier: NCT02065336	Interventional, randomized, parallel assignment study to evaluate the efficacy and safety of ARC-520 in single or multiple doses vs. entecavir, chlorpheniramine, or placebo.	Strong reduction in HBsAg levels in patients HBeAg positive and naïve.	Influenza-like symptoms, pyrexia, rash. This study was terminated for regulatory and business reasons, due to the results of a non-clinical toxicology study.
ARB-1467	ClinicalTrials.gov Identifier: NCT02631096	Interventional, randomized, single-blinded, placebo-controlled study.	All 12 HBeAg-negative chronically infected patients experienced reductions in serum HBsAg levels, with an average reduction in serum HBsAg of 1.4 log_10_	No serious adverse events, ALT values remained normal
VIR-2218	ClinicalTrials.gov Identifier: NCT03672188	Interventional, randomized, quadruple masking, placebo-controlled study	Not available	Not available
	Clinical-Trials.gov Identifier: NCT02826018	Interventional, randomized, single-blind, placebo-controlled study	Not available	Not available
JNJ-3989	ClinicalTrials.gov Identifier: NCT03982186	Interventional, multi-center, double-blind, active-controlled, randomized study	Not available	Not available
	ClinicalTrials.gov Identifier: NCT04129554	Interventional, randomized, double blind, placebo-controlled study	Not available	Not available
**Core Protein inhibitors (class)**	**Reference**	**Study type**	**Results**	**Adverse events, limitations/comment**
NVR 3-778 (I)	Yuen et al., Gastroenterology, 2019.	Interventional, randomized, single-group assignment, phase 1 study	HBV DNA and HBV RNA reduction were observed in patients receiving >1200 mg/day NVR 3-778. No significative reduction in HBsAg levels	Fatigue, influenza-like symptoms, eczema. Viral rebound was observed after cessation of therapy.
JNJ-6379 (II)	ClinicalTrials.gov Identifier: NCT04667104	Interventional, phase-2, open-label, single-arm, multi-center study	Not available	Not available
	ClinicalTrials.gov Identifier: NCT04439539	Interventional, randomized, randomized, open-label, multi-center study	Not available	Not available
	ClinicalTrials.gov Identifier: NCT03361956	Interventional, phase 2a, randomized, partially blind, placebo-controlled study	Not available	Not available
RO7049389 (I)	ClinicalTrials.gov Identifier: NCT02952924	Interventional, randomized, double masking, parallel assignment study	HBV DNA and HBV RNA decline observed across all cohorts	HBsAg did not change significantly. Small study population and brief duration of treatment
ABI-H0731 (II)	ClinicalTrials.gov Identifier: NCT03577171	Interventional, randomized, triple blind, placebo-controlled, multi-center study	Combination of ABI-H0731 and ETV determined greater decline of HBV DNA and HBV RNA	Rush, influenza-like symptoms. Small study population
	ClinicalTrials.gov Identifier: NCT03576066	Interventional, randomized, double blind, placebo-controlled, multi-center study	100% of HBeAg negative patients had suppressed HBV DNA and pgRNA < 20	Nausea, diarrhea, upper respiratory tract infection. Small study population
	ClinicalTrials.gov Identifier: NCT04781647	Interventional, randomized, open-label, multi-center study	Not available	Not available
	ClinicalTrials.gov Identifier: NCT04454567	Interventional, randomized, double blind, multi-center study	Not available	Not available
GLS4 (I)	ClinicalTrials.gov Identifier: NCT04147208	Interventional, randomized, open-label, multi-center study	Not available	Not available
	ClinicalTrials.gov Identifier: NCT04551261	Interventional, non-randomized, open-label, multi-center study	Decline in level of HBV DNA were between −1.42 and −3.5 log_10_ IU/mL	ALT, AST, and GGT elevation
ABI-H2158	ClinicalTrials.gov Identifier: NCT03714152	Interventional, randomized, triple blinded, placebo-controlled, multi-center study	HBV DNA mean change from baseline were −2.3, −2.5, and −2.7 log_10_ IU/mL in patients receiving ABI-H2158 100 mg, 300 mg, or 500 mg, respectively, for 2 weeks	Hypertriglyceridemia, dyspepsia, insomnia, elevated ALT
	ClinicalTrials.gov Identifier: NCT04398134	Interventional, randomized, double blinded, placebo-controlled, multi-center study	Not available	Not available
JNJ-0440	ClinicalTrials.gov Identifier: NCT03439488	Interventional, randomized, double blinded, placebo-controlled study.	Not available	Not available
**Novel Polymerase Inhibitors**	**Reference**	**Study type**	**Results**	**Adverse events, limitations/comment**
Tenofovir Exalidex	ClinicalTrials.gov Identifier: NCT03279146	Phase 1, interventional, open-label, part randomized study	Not available	Not available
Besifovir	ClinicalTrials.gov Identifier: NCT04202536	Interventional, randomized, open-label, multi-center study	Not available	Not available
	ClinicalTrials.gov Identifier: NCT02792088	Interventional, randomized, double blinded, multi-center clinical trial	Not available	Not available
	Do Seon Song, et al., Clin Mol Hepatol, 2021	Interventional, randomized, double blinded, multi-center clinical trial. There were two treatment arms: Besifovir (150 mg QD) + L-carnitine (660 mg QD) + placebo vs. TDF (300 mg QD) + placebo + placebo for the first 48 weeks then both arms received Besifovir (150 mg QD) + L-carnitine (660 mg QD) for 48 weeks	Virological response rates were similar in the two arms.	Bone mineral density and renal function were well-preserved
	Yeon Woo Jung et al., J Korean Med Sci, 2020	Interventional, randomized, open-label, multi-center clinical trial	BSV + carnitine, as compared to ETV or TDF, was not associated with hepatic steatosis improvement	Small study population (24 vs. 251), small follow-up period (6 months)
	ClinicalTrials.gov Identifier: NCT04249908	Interventional, non-randomized, open-label to assess pharmacokinetics in impaired renal function	Not available	Not available

**Table 2 viruses-13-01767-t002:** Some of the main immunomodulatory drugs under investigation in the HBV chronic infection field. TLR: toll-like receptor; CHB: chronic hepatitis B; ISG15: interferon stimulating gene 15; HBsAg: hepatitis B surface antigen; HBeAg: hepatitis B early antigen; HBV: hepatitis B virus; DNA: deoxyribonucleic acid; NAs: nucleo(s)tide analogues; TAF: tenofovir alafenamide fumarate; PD: programmed death cell protein; ALT: alanine aminotransferase; GI: gastro-intestinal; FxR: farnesoid X receptor.

Drugs	Reference	Study Type	Results	Adverse Events/Limits
VESATOLIMOD (GS-9620) TLR7 agonist	Janssen HLA, J Hepatol. 2018; 68(3):431–440.	Phase II, multi-center, double-blind, randomized, placebo-controlled trial on the safety, efficacy, and pharmacodynamics of vesatolimod in CHB patients	No significant differences in HBsAg levels were observed among the different groups.	Fatigue, nausea, headache
	Agarwal K, J Viral Hepat. 2018;25(11):1331–1340.	Phase II, multi-center, randomized, double-blind, placebo-controlled study.	No significative differences in HBsAg changes from baseline were observed. HBV DNA suppression rates were similar among the different groups.	Chills, fatigue, nausea, headache.
RO7020531 TLR7 agonist	ClinicalTrials.gov Identifier: *NCT02956850*	Phase I multi-center, randomized, placebo-controlled study to evaluate the safety, tolerability, pharmacokinetics, and pharmacodynamics of RO7020531 in healthy volunteers and CHB patients	Not available	Not available
RO6864018 (RG7795) TLR7 agonist	ClinicalTrials.gov Identifier: *NCT02391805*	Phase II multi-center, randomized, partially double blind, placebo-controlled study to assess the safety, tolerability, pharmaco-kinetics/dynamics, and antiviral effect in virologically suppressed HCB patients	Not available	Not available
JNJ-4964 (AL-034/TQ-A3334) TLR7 agonist	ClinicalTrials.gov Identifier: *NCT04180150*	Phase II randomized, double-blind, placebo-controlled study to evaluate the safety and efficacy of the drug in virologically suppressed or not CHB patients	Not available	Not available
SELGANTOLIMOD (GS9688) TLR8 agonist	Gane EJ, Hepatology, https://doi.org/10.1002/hep.31795	Phase Ib, randomized, placebo-control study in viremic and virally suppressed CHB patients, in order to assess the safety and tolerability of the drug	Selgantolimod was safe and tolerated well in both viremic and virally suppressed CHB patients	Headache, nausea, and dizziness
	ClinicalTrials.gov Identifier: *NCT03615066*	Phase II multi-center, double-blind, randomized, placebo-controlled trial to evaluate the safety, tolerability, and anti-viral activity of selgantolimod in CHB naïve patients	No differences in HBsAg levels or HBV DNA	Not available
	ClinicalTrials.gov Identifier: *NCT03491553*	Phase II multi-center, double-blind, randomized, placebo- controlled trial to evaluate safety, tolerability, and anti-viral activity of selgantolimod in CHB virologically suppressed patients	HBsAg negativization in 10–22% of the patients; a minority of HBeAg positive subjects obtained seroconversion	Not available
NIVOLUMAB PD-1 inhibitor	Gane E, J Hepatol. 2019; 71(5):900–907.	Phase I single center, open-label clinical trial assessing the safety and efficacy of nivolumab with or without GS-4774 (a yeast based therapeutic vaccine) in CHB patients	Declines of HBsAg levels were more evident in the association therapy groups than with Nivolumab alone.	Fatigue, cough, headache increase of ALT levels.
CEMIPLIMAB Anti PD-1 monoclonal antibody	ClinicalTrials.gov Identifier: *NCT04046107*	Phase I–II ascending multiple doses, open-label, non-randomized clinical trial assessing the safety and efficacy of cemiplimab in CHB virally suppressed patients.	Not available	Not available
ENVAFOLIMAB (ASC-22) PD-L1 single domain antibody Fc fusion	ClinicalTrials.gov Identifier: *NCT04465890*	Phase II randomized, parallel assignment study to evaluate the safety, tolerability, and efficacy of ASC-22 in CHB HBeAg-negative, already treated with NAs patients.	Not available	The phase IIa reported only grade 1 AE, highlighting the drug tolerability. IIb not available.
HLX10 Anti PD-1 humanized monoclonal antibody	ClinicalTrials.gov Identifier: *NCT04133259*	Phase II multi-center, open-label clinical trial.	Not available	Not available
EYP001a FxR agonist	Erken R, J Hepatol, 2018. 68(1):S488-S489	Phase I, open-label, randomized, 4-way cross-over study in subjects with chronic HBV infection, in order to assess the pharmacokinetics (fasted/fed), safety, tolerability, and pharmacodynamics of single oral doses of EYP001	Oral doses of EYP001a were well-tolerated and induced a long-lasting engagement of FxR. HBV markers were not reduced significantly after single oral doses, but an in silico model showed an inhibitory effect of the drug on HBV replication with prolonged treatment.	Mild, short-lasting GI AE.
	ClinicalTrials.gov Identifier: *NCT04365933*	Phase IIa multi-center, randomized open-label experimental study to assess EYP001a safety and antiviral effect in non-treated CHB patients in combination with ETV and pegylated IFN alpha2a	Not available	Not available
APG-1387 Inhibitor of apoptosis protein (IAP) inhibitor	ClinicalTrials.gov Identifier: *NCT04568265*	Phase II multi-center, open-label study to evaluate the safety, tolerability, pharmacokinetic profile, and preliminary anti-HBV efficacy of APG-1387 in combination with entecavir, and to determine its optimal dose	Not available	Not available

**Table 3 viruses-13-01767-t003:** Main clinical trials evaluating therapeutic vaccines employed in HBV chronic infection therapies. HBsAg: hepatitis B surface antigen; HBcAg: hepatitis B core antigen; CHB: chronic hepatitis B; ETV: entecavir; NA, NUC: nucleos(t)ide analogue; HBV: hepatitis B virus; DNA: deoxyribonucleic acid; HBeAg: hepatitis B early antigen; HBx: hepatitis B virus x protein; IL-2: interleukin-2; IL-12: interleukin-12; ALT: alanine aminotransferase; CD4+: cluster of differentiation 4+; CD8+: cluster of differentiation 8+; INF-γ: interferon-γ; TGF-β: transforming growth factor-β; TNF: tumor necrosis factor; T-regs: regulatory T-cells.

Name Vaccine Type	References	Study Design	Results	Adverse Events
DV-601 (Theravax) Protein-based vaccine containing HBsAg and HBcAg plus an adjuvant	Spellman. The 21st conference of the Asian Pacific association for the study of the liver: oral presentations 17 February, 2011 (Thursday). Hepatol Int. 2011; 5(1):3–558, doi:10.1007/s12072-010-9241-z	Phase I dose-escalation study in treatment-naïve or treatment-tolerant adult patients with CHB, in association with ETV.	All patients showed reductions in HBV DNA, s and e antigens, and HBV-specific lymphoproliferative response; antibodies to s and e antigens developed in the higher dose groups.	Transient chills, malaise, fatigue, and headache
	ClinicalTrials.gov Identifier: NCT01023230	Phase I b dose-escalation study to assess the safety and tolerability in subjects with CHB on concurrent treatment with NA.	Not available.	Not available
HepTcell ™ Mix of nine synthetic peptides comprising HBV specific T-cell epitopes	Lim, Young-Suk, J Hepatol 70.1 (2019): e50–e51.	Phase Ib study for evaluation of Hep T-cell HBV-specific immunotherapy in NA-controlled, HBeAg negative CHB	Hep T-cell immunotherapy was well-tolerated, but had no effects on HBsAg levels. It also showed HBV-specific immune responses.	No adverse events reported
	ClinicalTrials.gov Identifier: NCT04684914	Phase II randomized, double blind, and placebo-controlled study of Hep T-cells in treatment of naïve inactive CHB patients with low HBsAg levels.	Not available.	Not available
εPA-44 Immunodominant B-cell epitope of PreS2 18–24 region, the CTL epitope of HBcAg18–27 and the universal T-helper epitope of tetanus toxoid 830–843	ClinicalTrials.gov Identifier: NCT01326546	Phase II randomized, double-blind, placebo-controlled, multi-center clinical trial to evaluate the efficacy and safety of εPA-44 in HBeAg-positive CHB patients.	No differences among the two groups, in terms of HBeAg seroconversion, at week 48.	Not available
	ClinicalTrials.gov Identifier: NCT00869778	Phase II multi-center, randomized, double-blind, placebo-controlled clinical trial to evaluate the efficacy and safety of εPA-44 in treating CHB patients.	HBeAg seroconversion was achieved at week 76 by 38.8% of patients in the higher dose group, 28.6% of subjects in the lower dose group, and 20.2% of patients in the placebo group.	Not available
	ClinicalTrials.gov Identifier: NCT02862106	Second stage of the phase II study mentioned above, open-label. Re-treatment of all subjects with partial or no response to experimental treatment.	Re-treatment with high dose εPA-44 did not affect HBeAg seroconversion.	Not available
ABX 203 (NASVAC) Combination therapeutic vaccine containing hepatitis B surface antigen and hepatitis B core antigen	Al Mahtab M, PLoS One. 2018; 22; 13(8):e0201236.	Phase III randomized open-label efficacy and comparative study of ABX 203 in treatment-naïve patients. Patients were randomized to receive ABX 203 or Peg-IFN	The experimental group achieved higher rates of HBeAg seroconversion and HBV DNA suppression as well as a lower progression to cirrhosis.	NASVAC-treated patients experienced ALT elevations more frequently than controls
	Wedemeyer, The international liver congress (EASL) 2017, 66(1):S101.	Phase IIb/III multi-center, randomized, open-label study on efficacy of ABX203 vaccine to maintain HBV viral suppression and liver enzymes normalization after NAs treatment interruption in HBeAg-negative CHB patients treated for at least two years with NAs	The experimental drug did not prevent viral relapse after NAs interruption, but Tenofovir-treated patients relapsed earlier than Entecavir-treated subjects.	Injection site reactions
INO-1800 + INO-9112 INO-1800 DNA vaccine encoding HBsAg and a consensus sequence of HBcAg INO-9112: DNA plasmid encoding human interleukin 12	ClinicalTrials.gov Identifier: NCT02431312;	Phase I, randomized, open-label, active-controlled dose escalation study on the safety, tolerability, and immunogenicity of INO-1800, alone or in combination with INO-9112, in NUC-treated CHB patients	Preliminary results show that INO-1800 is safe and well-tolerated, and determines a virus-specific T-cell immune response.	Not available
JNJ-64300535 DNA vaccine	ClinicalTrials.gov Identifier: NCT03463369	Phase I, double-blind, randomized, placebo-controlled study to evaluate the safety, tolerability, reactogenicity, and immunogenicity of JNJ-64300535	Not available	Not available
GS-4774 Recombinant yeast-based vaccine containing HBsAg, HBcAg, and HBx proteins	Lok AS. J Hepatol. 2016 Sep; 65(3):509–16	Phase II, randomized, open-label study on the safety and efficacy of GS-4774 for the treatment of virally suppressed CHB patients.	No clinical benefits in HBsAg clearance, even if, in the higher dose group, three subjects experienced a >0.5 log_10_IU/mL reduction in HBsAg levels; five patients in the experimental group obtained HBeAg seroconversion, none in the control group.	Injection site reactions
	Boni C, Gastroenterology. 2019 Jul;157(1):227–241.e7.	Phase II, randomized open label multi-center study on the safety and efficacy of GS-4774 plus tenofovir disoproxil fumarate (TDF) in CHB.	Adding GS-4774 to TDF did not affect HBsAg blood levels, but determined the production of IFN, TNF, and IL2 by CD8+ T-cells.	Headache, myalgia, fatigue, and site injection reactions
TG-1050 Adenovirus 5-based therapeutic vaccine expressing core, polymerase, and surface antigen HBV proteins	Zoulim F. Hum Vaccin Immunother. 2020; 16(2):388–399.	Phase I, double-blind, randomized, placebo-controlled multi-cohort study in CHB patients in treatment with NAs	TG-1050 showed a good safety profile and induced an HBV-specific cellular immune response. HBsAg serum concentration did not decrease significantly.	Injection site reactions
HB-110 Second generation adenoviral-based DNA vaccine encoding for S, L, core and polymerase protein, adjuvated with IL-12	Yoon SK. Liver Int. 2015 Mar; 35(3):805–15.	Phase I, single-center, randomized, open-label, dose escalating study on the safety of HB-110 combined with oral adefovir in CHB	HB-110 exhibited positive effects on ALT normalization and maintenance of HBeAg seroconversion.	Headache and fatigue
YIC (Yeast-derived immune complexes) HBsAg–hepatitis B immunoglobulin (HBIG) complex with alum as adjuvant	Xu DZ. PLoS One. 2008 Jul 2; 3(7):e2565.	Phase II randomized controlled clinical trial in HBeAg positive CHB patients. Six doses of YIC or alum as placebo	60 microgr YIC showed better results than placebo in HBeAg seroconversion (21.8% vs. 9%).	Pain in the injection site, pruritus, and swelling
	Xu DZ. J Hepatol. 2013 Sep; 59(3):450–6.	Phase III randomized controlled clinical trial on CHB patients. Twelve doses of YIC or alum as placebo	HBeAb seroconversion rates were major for the placebo group (21.9% vs. 14%); decrease in serum HBV DNA and normalization of liver function were similar in both groups.	Transient ALT flares
	Zhou C. Hum Vaccin Immunother. 2017 Sep 2; 13(9):1989–1996.	Phase I randomized controlled clinical trial to assess YIC immunological mechanisms.	YIC determined increased CD4+ and CD8+ responses, a major production of INF-γ, and a decreased concentration of inhibitory cytokines (IL-10, TGF-β) and T-regs.	Not reported

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
