# Peer review of "Treatments for HBV: A Glimpse into the Future"

_viruses, 2021, doi:10.3390/v13091767_

Round 1

Reviewer 1 Report

This is an updated summary of current novel drug therapies for chronic hepatitis B. While details about the distinct molecules and approaches are well described, there is a poor assessment and overall evaluation of the prospects of the distinct treatments. A final section sub-heading discussing combination treatments and sequential therapies will be much appreciated.

Author Response

Dear reviewers,

We really appreciated your fine work, thank you for your valuable efforts. We totally understood the criticisms highlighted and in particular we have revised the English language helped by English native speakers. We are sorry for the tables configuration, in the original manuscript everything was clearly visible and readable. We must thank you not only for the formal corrections but especially for those concerning the content. We have made the required changes, revised the references and added more detail about the results of the different clinical trials testing the new anti-viral drugs both in animal and human studies. We really thank you again for all your efforts.

We thank the Reviewer for the gentle suggestions and comments. We completed the conclusion paragraph with a section dedicated to illustrate the possibility of combination treatments and sequential therapies, as suggested. We hope this version of our manuscript will be fully readable and complete. Thanks again.

Reviewer 2 Report

The manuscript by Bartoli et al. is a comprehensive review focusing on the newest drug developments for the treatment of chronic infection with the hepatitis B virus. The extensive listing and detailed description of various drug categories are interesting for the audience of “Viruses” and will provide a helpful resource for scientists in the field.

General comments:

The manuscript would immensely benefit from an improvement of the English language.

Some terms need to be replaced by the correct English word; e.g., title: a glimpse into the future; lines 25 and 76: desossiribonucleic; lines 71, 81, 86, and 210: resume(s), line 38: genic; Table 1, NVR 3-778 (I): mg/die; line 97: HBV l-surface protein; line 122 analogue nucleos(t)ide drugs; line 300, association; line 379: codifying; line 356: shew; and lines 477, 522, 542: bound.

Some sentences appear incomplete; e.g., lines 38-39: We excluded articles with not available full text…; lines 80-81: We will now review the principal novel therapies in phase of development for the treatment of HBV infection; and lines 118-119: GalNAc siRNA are uptaken by asialoglycoprotein receptor (ASGPR) on hepatocytes, resulting liver-specific [28].

Other sentences are hard to understand as written; e.g., lines 112-113: Artificial sequences are synthetized to reprogram silencing in different step of miRNA biosynthesis pathways; lines 139-140: HBc consists in a single 183-185 amino acids chain with a N-terminal forming the capsid assembly domain and a C-terminal forming a nucleic acid domain [32]; lines 224-225: TLRs ligands have been tested in vitro and in vivo animal models and TLR7 and TLR8 ligands resulted the most efficacious; lines 291-292: From in vitro studies, the most inhibitory receptor among the whole category is PD-1.; lines 293-295: …while intrahepatic HBV specific T cells are usually more exhausted and to be restored, an association with other inhibitory receptors blockade could be useful [59]; lines 298-299: ..in HBeAg negative HBV infected blood cells [60]; lines 397-398: …AIC649 underwent phase I trial showing an increase in IL-1β, IL-6, IL-8 and INF-γ amounts, reducing instead IL-10 levels and resulting well tolerated [76]; lines 454-456: Since they are monomers cutting single strand DNA, to determine a double stranded break, ZFs need the monomers dimerization and juxtaposition on the target sequence [85]; lines 481-482: A next generation sequencing map of CRISPR/Cas9 induced viral cccDNA mutations revealed that the majority of indels resulted as small deletions [90]; and lines 542-544: HBV core protein is a structural protein in cccDNA and its bound determines an alteration reducing cccDNA nucleosomal spacing, regulating its transcriptional activity.

Specific comments:

Line 42: Better to state Hepatitis B Virus Replication Cycle instead of Hepatitis B Replicative Cycle.

Line 48: Then, the nucleocapsid is released into the cell…Since viral particles are internalized via NTCP, it would be better to state cytoplasm instead of cell.

Line 78: HBeAg is an abbreviation for the HBV early or e antigen not the hepatitis B envelope antigen

Lines 85-86: What is meant by that new drugs (blockers of entry receptor, RNA interference, core protein inhibitors, novel polymerase inhibitors and ribonuclease H (RNAseH) inhibitors) implement the already existing ones?

Table 1 and Table 2, The right column describing adverse events/limitations cannot be read in the pdf file, please change the width. The tables are pretty large and should be shorten by removing trials without available data or reported adverse events. These trials could be mentioned in the text.

Table 1, Bulevirtide: What is meant by that the HBsAg response was statistically reached at 48 weeks in patients receiving Myr 2 mg + PEG INF? Compared to Mry 2 or PEG INF alone? Please state the difference in HBsAg reduction.

Table 1, Besifovir, Do Seon Song, et al.: What is meant by in the 2 arms BSV-BSV vs TDF vs BSV? Should this read BSV-TDF vs TDF vs BSV?

Line 102: Trial identifier numbers in the text do not match with those listed in Table 1.

Lines 132-134, 177-183, 184-189, and 190-196: Table 1 states that no data are available or adverse events were reported for ARB-1467, GLS4, ABI-H2158, and JNJ-0440, while the text provides results for all drugs and additionally states that the study was terminated (ARB-1467) or that serious ALT flares were observed (GSL4).

Lines 135-137: NCT02826018 is not listed in Table 1 for VIR-2218.

Lines 140-141: Please explain what human and viral proteins need to be engaged by HBc for creating a firm, protective capsid.

Line 155: Should probably read HBV DNA instead of HBD-DNA. What was the reduction in DNA and RNA with NVR 3-778?

Line 157: Please state if JNJ-6379 is a class I or II CpAm.

Line 164: Please state what kind of compounds JNJ-3989 and JNJ-6379 are.

Line 172: Has a depletion of the cccDNA pool really been shown for ABI-H0731?

Lane 182: A better word is experienced instead of experimented.

Lines 200-201. Please state if Tenofovir exalidex reduces HBsAg and cccDNA or not. It “seems to reduce” is not very scientific.

Line 206: Immunotherapeutic approaches targeting the different alterations are now reviewed. Please state these alterations.

Line 210: What is meant by replacing the host T-cells response via the reported drugs?

Table 2 and lines 251-261: All trials with Inarigivir/SB 9200 have been terminated because hepatic adverse events occurred and one patient developed a progressive liver failure and died from multi-organ failure. No clinical trial for testing this compound is ongoing.

Table 2: Nivolumab is a monoclonal antibody that binds PD-1. Please state what kind of compound GS-4774 is.

Table 2: Please define IAP inhibitor.

Many (animal) study results described under 4.1 Immunotherapy cite references 46 or 51, both are recent review articles. However, it would be more accurate to cite the original papers if referring to the results of these studies; e.g., lines 225-227, 271-275, 237-238, and 277-280.

Lines 229-231: The statement that Vesatolimod/GS-9620 had no effect on cccDNA in animal models is incorrect.

Lines 233-236: This statement should be moved up in the paragraph as these are the results obtained with Vesatolimod/GS-9620 and not with RO7020531 as implied

Lines 237-241: Well-tolerated is a better word for good-tolerated. Should read in vivo as the results on GS-9688/Selgantolimod refer to studies in patients.

Line 243: Please explain what an intracytoplasmic PAMP receptor is.

Line 254: Reference 54 is incorrect.

Lines 263-269: Please state that DMXXAA caused adverse events as suspected by declines in body weight.

Line 286 and 290: Better to refer to antibodies or inhibitors than naming these compounds PD-1/PD-L1 agonists.

Line 300-304: Reference 61 is not the original paper for this study.

Lines 310-319: This paragraph needs improvement as it is not clear what the authors try to state.

Table 3: It might be intentionally but the design of this table is different to those of Tables 1 and 2. The right column listing references cannot be read in the pdf file, please change the width.

Table 3: Is it true that the results for ePA-44 and pCMVS2.S are identical? Under both compounds it is stated that HBeAg seroconversion was achieved at week 76 by 38.8% of patients from higher dose group, 28.6% of subjects from the lower dose group and 20.2% of patients from placebo group.

Table 3, GS-4774: Which tenofovir (TDF or TAF) was used in the trial? What is meant by no differences in two group for what concerns HBsAg levels?

Table 3, TG-1015 should read TG-1050.

Line 371-372: It appears that monocytes instead of myocytes were meant as antigen presenting cells.

Line 429-432: Reference 64 is not the original paper for this study.

Lines 467-470: In addition to cccDNA reductions, was an effect on secreted virions and HBsAg observed? If so, please include.

Lines 492-494: Please explain what further advances are needed.

Line 502: Should read cytoplasm instead of cytoplasmic. Please rephrase this sentence.

Lines 533-535: Please state what good results have been obtained in preliminary studies.

Lines 568-572: Please state what good results on safety and viral replication were obtained.

Line 582: NCT04365933 is not listed in Table 2 under studies testing FXR agonists.

Lines 583-591: Are there differences in the mechanism of action of the IAP inhibitor APG-1387 (Table 2) and the cIAP inhibitor Birinapant?

Author Response

Dear reviewers,

We really appreciated your fine work, thank you for your valuable efforts. We totally understood the criticisms highlighted and in particular we have revised the English language helped by English native speakers. We are sorry for the tables configuration, in the original manuscript everything was clearly visible and readable. We must thank you not only for the formal corrections but especially for those concerning the content. We have made the required changes, revised the references and added more detail about the results of the different clinical trials testing the new anti-viral drugs both in animal and human studies. We really thank you again for all your efforts.

Reviewer 2

We sincerely thank the Reviewer for the huge, accurate and valuable revision work done. The suggestions received were precious. We are sorry we were not clear in English writing, that’s why the entire manuscript has been revised by an English native speaker. We hope that this version of the manuscript will be clearer and fully understandable.

Here we report a point-by-point response to the questions and suggestions expressed by reviewer nr 2:

  • Line 42 (original paper): Better to state Hepatitis B Virus Replication Cycle instead of Hepatitis B Replicative Cycle.

Answer: Line 38. We corrected it.

  • Line 48 (original paper): Then, the nucleocapsid is released into the cell…Since viral particles are internalized via NTCP, it would be better to state cytoplasm instead of cell.

Answer: Line 43. We made the suggested correction.

  • Line 78 (original paper): HBeAg is an abbreviation for the HBV early or e antigen not the hepatitis B envelope antigen

Answer: Line 70 (revised paper): corrected

  • Lines 85-86: What is meant by that new drugs (blockers of entry receptor, RNA interference, core protein inhibitors, novel polymerase inhibitors and ribonuclease H (RNAseH) inhibitors) implement the already existing ones?

Answer: Lines 73-74. It means that these drugs can expand the therapeutic options that already exist. The sentence was changed in the text.

  • Table 1 and Table 2, The right column describing adverse events/limitations cannot be read in the pdf file, please change the width. The tables are pretty large and should be shorten by removing trials without available data or reported adverse events. These trials could be mentioned in the text

Answer: in the original document, in word format, all column were of adequate width, when the document was uploaded. We reported the problem to the editorial service in order to avoid further layout problems in conversion doc. to pdf. We preferred to keep all the studies in the tables so that it will be easier to find them.

  • Table 1: What is meant by that the HBsAg response was statistically reached at 48 weeks in patients receiving Myr 2 mg + PEG INF? Compared to Mry 2 or PEG INF alone? Please state the difference in HBsAg reduction

Answer: Table 1, bulevirtide.   We have better explained the results in the table. The authors did not report other details about the difference in HBsAg reduction between the groups.

  • Table 1 Besifovir, Do Seon Song, et al.: What is meant by in the 2 arms BSV-BSV vs TDF vs BSV? Should this read BSV-TDF vs TDF vs BSV?

Answer: Table 1, Besifovir, Do Seon Song, et al. We better explained in the table, illustrating the arms of treatment.

  • Line 102 original document: Trial identifier numbers in the text do not match with those listed in Table 1

Answer: Line 89. NCT03852719 and NCT03852433 were not reported in table 1 because they refer to HDV therapy clinical trials. They were only cited in the text.

  • Lines 132-134, 177-183, 184-189, and 190-196 original document: Table 1 states that no data are available or adverse events were reported for ARB-1467, GLS4, ABI-H2158, and JNJ-0440, while the text provides results for all drugs and additionally states that the study was terminated (ARB-1467) or that serious ALT flares were observed (GSL4).

Answer: Lines120-124, 171-176, 177-184, 185-190 and table 1. We fixed the mistakes; for JNJ-0440 no data is available about adverse events.

  • NCT02826018 is not listed in Table 1 for VIR-2218.

Answer: Table 1. NCT02826018 is now listed in Table 1

  • Lines 140-141 original paper: Please explain what human and viral proteins need to be engaged by HBc for creating a firm, protective capsid.

Answer: Lines 133-136. We explained it in the text.

  • Line 155 original paper: Should probably read HBV DNA instead of HBD-DNA. What was the reduction in DNA and RNA with NVR 3-778?

Answer: Lines 143-150. We corrected and explained it in the text.

  • Line 157 original paper: Please state if JNJ-6379 is a class I or II CpAm

Answer: Line 151.We explained it as suggested.

  • Line 164 original paper: Please state what kind of compounds JNJ-3989 and JNJ-6379 are

Answer: Lines 116-119, 151. We explained it as suggested. 

  • Line 164 original paper: Has a depletion of the cccDNA pool really been shown for ABI-H0731?

Answer: Lines 150-170. We have explained it in the text.

  • Line 182 original paper: A better word is experienced instead of experimented.

Answer: Line 182. We changed it as suggested.

  • Line 200-201 original paper: Please state if Tenofovir exalidex reduces HBsAg and cccDNA or not. It “seems to reduce” is not very scientific

Answer: Lines 195-196. We corrected it.

  • Line 206 original paper: Immunotherapeutic approaches targeting the different alterations are now reviewed. Please state these alterations.

Answer: Lines 203-209. The sentence was rephrased.

  • Line 210 original paper: What is meant by replacing the host T-cells response via the reported drugs?

Answer: Lines 205-209. The sentence was rephrased.

  • Table 2 and lines 251-261 original paper: All trials with Inarigivir/SB 9200 have been terminated because hepatic adverse events occurred and one patient developed a progressive liver failure and died from multi-organ failure. No clinical trial for testing this compound is ongoing

Answer: Lines 256-257. We really thank the reviewer since we made an important mistake not realizing that this drug was withdrawn from clinical trials. We corrected the chapter and deleted the sections of table 2 reporting clinical trials about Inarigivir. 

  • Table 2: Nivolumab is a monoclonal antibody that binds PD-1. Please state what kind of compound GS-4774 is.

Answer: The information has been added in table 2 (a yeast based therapeutic vaccine)

  • Table 2: Please define IAP inhibitor.

Answer: The definition was added in table 2 (inhibitor of apoptosis protein).

  • Many (animal) study results described under 4.1 Immunotherapy cite references 46 or 51, both are recent review articles. However, it would be more accurate to cite the original papers if referring to the results of these studies; e.g., lines 225-227, 271-275, 237-238, and 277-280 (original paper).

Answer: The correct citations have been added. Lines 225-227; 235-236;261-264; 270-273; 276-278.

  • Lines 229-231 original paper: The statement that Vesatolimod/GS-9620 had no effect on cccDNA in animal models is incorrect.

Answer: Lines 229-230. The sentence was corrected.

  • Lines 233-236 original paper: This statement should be moved up in the paragraph as these are the results obtained with Vesatolimod/GS-9620 and not with RO7020531 as implied

Answer: Lines 228-234. The sentence was better formulated.

  • Lines 237-241 original paper: Well-tolerated is a better word for good-tolerated. Should read in vivo as the results on GS-9688/Selgantolimod refer to studies in patients.

Answer: Lines 235-239. We modified following the gentle suggestion.

  • Line 243 original paper: Please explain what an intracytoplasmic PAMP receptor is

Answer: The acronym was already explained at lines 221-222.

  • Line 254 original paper: Reference 54 is incorrect.

 Answer Lines 249-251, Reference 58. The reference was replaced with the correct one: “Suresh M, Korolowicz KE, et al. Antiviral Efficacy and Host Immune Response Induction during Sequential Treatment with SB 9200 Followed by Entecavir in Woodchucks. PLoS One. 2017 Jan 5;12(1):e0169631”.

  • Lines 263-269 original paper: Please state that DMXXAA caused adverse events as suspected by declines in body weight.

Answer Lines 266-267. The information was added.

  • Line 286 and 290 original paper: Better to refer to antibodies or inhibitors than naming these compounds PD-1/PD-L1 agonists.

Answer Lines 286 and 289. We corrected with the term inhibitors.

  • Line 300-304 original paper: Reference 61 is not the original paper for this study.

Answer Lines 298-301. Reference nr 69. The correct citation was added: “Liu J, Zhang E et al. Enhancing virus-specific immunity in vivo by combining therapeutic vaccination and PD-L1 blockade in chronic hepadnaviral infection. PLoS Pathog. 2014 Jan;10(1):e1003856.

  • Lines 310-319 original paper: This paragraph needs improvement as it is not clear what the authors try to state.

Answer Lines 307-314: we made changes in order to make it more understandable.

  • Table 3: It might be intentionally but the design of this table is different to those of Tables 1 and 2. The right column listing references cannot be read in the pdf file, please change the width.

Answer Table 3 was re-designed in order to be identical to the others; the version sent to the paper was perfectly readable, there might have been a configuration error once uploaded.

  • Table 3: Is it true that the results for ePA-44 and pCMVS2.S are identical? Under both compounds it is stated that HBeAg seroconversion was achieved at week 76 by 38.8% of patients from higher dose group, 28.6% of subjects from the lower dose group and 20.2% of patients from placebo group.

Answer:  Table 3: No, it was a mistake. The reported results were those of ePA-44. Everything has been corrected in table 3. Since that of pCMVS2.S, in the original paper, is an old clinical trial, in this last version of the manuscript it has been removed. 

  • Table 3, GS-4774: Which tenofovir (TDF or TAF) was used in the trial? What is meant by no differences in two group for what concerns HBsAg levels?

Answer: Table 3.TDF was used in the clinical trial. The sentence has been rephrased and the results section updated.

  • Table 3, TG-1015 should read TG-1050.

Answer Table 3: We corrected the drug name.

  • Line 371-372 original paper: It appears that monocytes instead of myocytes were meant as antigen presenting cells.

Answer: Lines 370-372: we were referring to local (site of vaccine injection) antigen presenting cells. Sentence rephrased.

  • Line 429-432 original paper: Reference 64 is not the original paper for this study.

Answer Lines 425-429. Citation nr 91.The correct citation was added: “Bohne F, Chmielewski M, et al. T cells redirected against hepatitis B virus surface proteins eliminate infected hepatocytes. Gastroenterology. 2008;134(1):239–47”.

  • Lines 467-470 original paper: In addition to cccDNA reductions, was an effect on secreted virions and HBsAg observed? If so, please include.

Answer Lines 460-464. The sentence was updated.

  • Lines 492-494 original paper: Please explain what further advances are needed.

Answer As reported in the text (lines 477-485) the central issues for this technology consist in the delivery strategy, the need of multiple single guide RNAs to have better results and the risk of off-target cleavage. These are the further advances we were referring to. In order not to be redundant we chose not to change the sentence.

  • Line 502 original paper: Should read cytoplasm instead of cytoplasmic. Please rephrase this sentence.

Answer Line 492. We rephrased the sentence.

  • Lines 533-535 original paper: Please state what good results have been obtained in preliminary studies.

Answer Lines 520-523.The results have been added.

  • Lines 568-572 original paper: Please state what good results on safety and viral replication were obtained.

Answer Lines 554-556. The details asked have been added.

  • Line 582 original paper: NCT04365933 is not listed in Table 2 under studies testing FXR agonists.

Answer Table 2: the study has been added.

  • Lines 583-591 original paper: Are there differences in the mechanism of action of the IAP inhibitor APG-1387 (Table 2) and the cIAP inhibitor Birinapant?

Answer No, these two molecules belong to the same class. They are both Smac-mimetics.

Reviewer 3 Report

The authors have done a great deal of work in searching and listing all the information, but the written presentation is very poor. The English is to be frank, awful, as exemplified by numerous spelling, preposition and plurality errors. The wrong words are used in some sentences. It must be rewritten by a native speaker to make it legible. It detracts the reader from the information provided, which was why I wished to review the paper, so that I could learn something about HBV therapies. I have corrected the English up to page 7, but as a reviewer this is not my job. This is the major reason as to why I have rejected the manuscript. I understand that as a non-native speaking English research group this is a hard task, but this is what is expected.

I do not understand why Hepatitis D was introduced into the text: page 3. How does it differ to HBV? The majority of the information I do not have a problem with, but in reading the information the reader becomes lost.

Minor points

  1. Line 22: 600,000
  2. Line 25: cccDNA equals covalently closed circular deoxyribonucleic acid
  3. Line 34: during the
  4. Line 55: instead of decades
  5. Line 58 and 59: as a template
  6. Line 62: polymerization, three
  7. Line 70: as an; Line 71: the cccDNA reservoir. Figure 1 illustrates….
  8. Line 80: phases of; Line 81; replace resume with “show”
  9. Line 86: replace resume with “lists”
  10. Line 98: The European….
  11. Line 108: on adequate binding…
  12. Line 112: reducing the
  13. Line 113: steps
  14. Line 118: taken up; Line 119: remove “resulting liver specific”
  15. Line 120: and caused a significant reduction
  16. Line 124: capable of interacting with…
  17. Line 126: replace evidenced with “shown”
  18. Line 132: LPN; in half of patients.
  19. Line 135: siRNA was investigated in a
  20. Line 139: acid; Line 145: genes; replace can with could.
  21. Line 152: replace formation with generation.
  22. Line 164: are now being recruited.
  23. Line 166: causes.
  24. Line 172: it was reported to decrease HBV…Line 175: of a combination protocol.
  25. Line 177: which can induce
  26. Line 180: levels
  27. Line 184: exhibited not experimented.

Author Response

Dear reviewers,

We really appreciated your fine work, thank you for your valuable efforts. We totally understood the criticisms highlighted and in particular we have revised the English language helped by English native speakers. We are sorry for the tables configuration, in the original manuscript everything was clearly visible and readable. We must thank you not only for the formal corrections but especially for those concerning the content. We have made the required changes, revised the references and added more detail about the results of the different clinical trials testing the new anti-viral drugs both in animal and human studies. We really thank you again for all your efforts.

Reviewer 3

We thank the Reviewer for his work, interest and suggestions. We hope this revised version will be more understandable and complete. Since HDV is the specific subject of another manuscript belonging to the same special issue we actively decided not to include it in our paper. We thank the reviewer again for his suggestion.

Round 2

Reviewer 1 Report

The authors addressed properly the comments risen by the referees. English editing has been extensive. Discussions and references have been updated.